# Intermolecular-Type Conical Intersections in Benzene Dimer

**DOI:** 10.3390/ijms24032906

**Published:** 2023-02-02

**Authors:** Attila Bende, Alex-Adrian Farcaş

**Affiliations:** National Institute for Research and Development of Isotopic and Molecular Technologies, Donat Street, No. 67-103, 400283 Cluj-Napoca, Romania

**Keywords:** TDDFT, spin-flipped TDDFT, CASSCF, intermolecular-type conical intersection, benzene dimer

## Abstract

The equilibrium and conical intersection geometries of the benzene dimer were computed in the framework of the conventional, linear-response time-dependent and spin-flipped time-dependent density functional theories (known as DFT, TDDFT and SF-TDDFT) as well as using the multiconfigurational complete active space self-consistent field (CASSCF) method considering the minimally augmented def2-TZVPP and the 6–31G(d,p) basis sets. It was found that the stacking distance between the benzene monomers decreases by about 0.5 Å in the first electronic excited state, due to the stronger intermolecular interaction energy, bringing the two monomers closer together. Intermolecular-type conical intersection (CI) geometries can be formed between the two benzene molecules, when (i) both monomer rings show planar deformation and (ii) weaker (approximately 1.6–1.8 Å long) C–C bonds are formed between the two monomers, with *parallel* and *antiparallel* orientation with respect to the monomer. These intermolecular-type CIs look energetically more favorable than dimeric CIs containing only one deformed monomer. The validity of the dimer-type CI geometries obtained by SF-TDDFT was confirmed by the CASSCF method. The nudged elastic band method used for finding the optimal relaxation path has confirmed both the accessibility of these intermolecular-type CIs and the possibility of the radiationless deactivation of the electronic excited states through these CI geometries. Although not as energetically favorable as the previous two CI geometries, there are other CI geometries characterized by the relative rotation of monomers at different angles around a vertical C–C axis.

## 1. Introduction

Internal conversion is one of the most important processes by which the electronic excited states induced in molecules by an electromagnetic radiation are rapidly and non-radiatively converted to the ground electronic state [1]. This phenomenon is strongly related to critical points on the potential energy surface, called a conical intersection (or CI) [2,3,4], defined as a crossing between states when a higher electronic state moves to a lower one. When a molecular system passes through a CI point, the Born–Oppenheimer approach becomes invalid and the coupling between electronic and nuclear motion becomes important [5,6,7]. The relaxation dynamics through these crossings often mean an ultra-fast transfer [7] of the higher electronic state to the lower one, typically on the femtosecond time scale.

Benzene is one of the simplest aromatic molecules, whose photophysical and photochemical behavior has been extensively studied by both theoretical and experimental methods [8,9,10,11,12,13,14,15,16,17,18,19,20]. Hence, one important feature of this molecule is that directly probing the S${}_{1}$/S${}_{0}$ radiative relaxation is almost impossible due to the very small transition probability from the ground to the first excited state and thus it is almost impossible to directly excite the first excited state. At the same time, using the time-resolved photoelectron spectroscopy (TRPES) technique, an ultra-fast internal conversion from the S${}_{2}$(π, π*) to the S${}_{1}$(π, π*) and then from S${}_{1}$ to S${}_{0}$ states was experimentally demonstrated [8,9,10]. The ultra-fast character of this relaxation pathway might be given by the relative proximity of the S${}_{1}$/S${}_{0}$ and S${}_{2}$/S${}_{1}$ crossing points [11,12]. However, in the methyl-substituted benzene derivatives (toluene and o-xylene) the relaxation time is significantly higher [10], due to the larger distance between the two critical points on the potential energy hyper-surface. Theoretical studies performed at the multiconfigurational CASSCF level of theory have shown that the radiationless relaxation pathway between the S${}_{1}$ and S${}_{0}$ states is mainly characterized by the well-known half-boat CI geometry, but ten other geometries can also be involved in the internal conversion process [13,14,15,16,17,18]. Furthermore, from an energetic point of view, the electron correlation plays an important role, since the conformational energy gap determined by the CI geometry is significantly reduced by the NEVPT2 method [19] compared to the CASSCF [20].

The benzene dimer also seems to be an interesting case study, as it can form the so-called excimer [21] structure in its excited state [22]. This is actually a system that exhibits an almost perfect parallel-stacking form (hexagonal prism) where the stacking distance is reduced by more than 0.5 Å compared to the ground-state configuration, due to the extra attractive interaction that occurs as a result of the excitation [23,24]. The benzene dimer, among other things, exhibits different photochemical behaviour compared to the monomer form. Namely, the spectrum of excited states shows an excimer state lower than the first excited state of the monomer benzene [25]. Furthermore, the mechanism of excimer formation in liquid benzene shows that this state is generated almost as soon as UV excitation occurs [26] due to an abundance of close-to-parallel benzene pairs. Non-adiabatic molecular dynamics study has shown that the excimer lifetime is in the range of tens of picoseconds [22], although the lack of broadening in the experimental excitation spectrum of fluorescence suggests that the lifetime may be much longer [27].

Another ab initio molecular dynamics study at the semi-classical (density functional tight-binding) theoretical level, where ultra-short laser pulses were explicitly taken into account, has shown that the bond distance between benzene monomers can be reduced to the value where the cyclobutane benzene dimer (called also syn-o,o’-dibenzene [28]) can be formed [29]. In this study, however, interesting geometric conformations that show CI character and govern both the photoinduced [2+2] cycloaddition reaction and the radiationless deactivation have been observed. These geometries were characterized not by the distortion of one monomer, but by the distortion of both monomers and the coupling between these monomers through a new covalent bond. It is important to note that these types of CI configurations are also important in the reaction of thymine photodimerization inside the DNA double helix [30,31,32]. Furthermore, similar intermolecular-type CI geometries were also found in the case of the benzene–uracil [33] complex or catechol dimer [34]. The main finding of the studies on catechol is that these intermolecular-type CIs are energetically competitive with monomeric-type CIs, having the half-boat form within the dimeric configuration [34].

Finding the location of the conical intersections on the potential energy hyper-surface and determining the corresponding critical geometry remains a serious task when the photochemical behavior of a molecule is revealed. It is necessary to choose both the appropriate theoretical model and the appropriate optimization algorithm [35,36,37,38,39,40]. Many small- and medium-sized molecules can be treated accurately by multiconfigurational approaches, but for larger molecules these studies can quickly become computationally expensive. In the latter cases, exploring the advantages offered by the DFT method can help to obtain results of a similar quality to those obtained for small- and medium-sized molecules. However, it should be noted that these advantages cannot always be exploited. For example, it is already known that time-dependent density functional theory (TDDFT) cannot accurately describe the degenerate electron states of geometries close to CIs due to the problem of the wrong dimensionality of the potential energy surface [41,42,43]. Levine et al. [41] proposed the so-called spin-flipped TDDFT method as a solution to overcome the aforementioned problem.

The main goal of the present investigation is to give a detailed description for the possible photoinduced radiationless decay channels in the benzene dimer over the newly proposed dimer-type conical intersection points considering the linear-response and spin-flipped TDDFT as well as the multiconfigurational CASSCF methods.

## 2. Results and Discussion

### 2.1. Equilibrium Geometries

The benzene dimer geometry with the lowest energy conformation shows the well-known T-shaped structure [44], but the parallel-shifted stacking configuration is energetically nearly as probable as the T-shaped structure [45,46], with only a very small difference in conformational energies (below the $k_{B}T$ thermal energy at room temperature) between the two structures. The ground-state equilibrium geometry structure of the benzene dimer was optimized considering the ωB97X-D3/ma-def2-TZVPP method. For the present case, the empirical dispersion-corrected DFT method gives the parallel-shifted stacking configuration as the most stable geometry (for the molecular graphics see Figure 1a). The computed intermolecular interaction energy between the monomers is −3.14 kcal/mol, the stacking distance is 3.58 Å, and both monomers keep the aromatic character of the C–C bonds (see the C${}_{i}$–C${}_{j}$ and C${}_{i}^{\prime }$–C${}_{j}^{\prime }$ bond length values in Table 1). These results are close to the values obtained with the higher correlation methods found in the literature [45,46]. As a next step, the optimized equilibrium geometry considering the first electronic excited-state level was computed considering the time-dependent DFT method, using the same exchange-correlation (XC) functional and basis set. The method ωB97X-D3/ma-def2-TZVPP showed very good behavior in obtaining the equilibrium geometry of the dopamine molecule in its first excited state [47]. The molecular structure of the S${}_{1}$ optimized geometry is shown in Figure 1b. Geometry optimization of the S${}_{1}$ excited-state level yields a more compact stacking configuration, where the parallel shift is also removed, and thus the dimer geometry shows an almost perfect parallel-stacking form. The energy of interaction between the monomers is significantly increased, the dimer dissociation energy being −14.00 kcal/mol. Accordingly, the stacking distance is also reduced, from 3.58 to 3.07 Å. The C${}_{i}$–C${}_{j}$ and C${}_{i}^{\prime }$–C${}_{j}^{\prime }$ bond lengths increase from 1.387 to 1.401 Å, but all carbon–carbon bonds remain equivalent, inducing a uniform enlargement of the hexagonal ring. Similar binding energy and C–C bond distance values were also found in Ref. [24], namely, 0.605 eV (13.95 kcal/mol) for the binding energy between monomers and 1.405 for C–C bond distances, when considering the CAM-B3LYP-D3/6-31+G(d) level of theory. Comparing the Mayer bond-order index (BOI), it is observed that, on average, the ground-state C–C BOIs for the S${}_{1}$ excited state decrease from 1.29 to 0.98, which implies a weakening of the bond strengths of the monomers, but at the same time the strength of the vertical C···C${}^{\prime }$ bonds increases, approaching a BOI of 0.1. The increase in the BOI values of the vertical bonds, even if this value is around 0.1, is due to the partial overlapping in the vertical direction of some of the monomer’s π molecular orbitals as shown in Figure 1 in Ref. [25]. As for the nature of the interaction energy between dimers, in the excited state, the energy is not just a balance of exchange repulsive and attractive dispersion energies as seen in the ground state, but a much more complex energy structure, e.g., strong electron–hole interactions induced by excitation, or more precisely the interaction of excitations localized on each of the two monomers and ion-pair (also called charge transfer) states. For more details see Ref. [24]. For the excited-state optimized dimer geometry, the interaction energy was calculated also by neglecting the D3 empirical dispersion correction. The resulting interaction energy was changed to −11.85 kcal/mol, a difference of 2.15 kcal/mol. This means that, although the deviation is not significant, it is still necessary to take into account dispersion-type electron correlation effects. What is more, we optimized the geometry of the dimer without taking dispersion effects into account at all. In this case, the stacking distance increased to 3.12 Å, which cannot be considered as a significant change (≈0.05 Å), but it is not negligible. Overall, it can be concluded that dynamic electron correlation effects should be taken into account at least for the estimation of the interaction energy.

### 2.2. Conical Intersection Geometries

It is well known that for a large number of molecules the decay of the excited state is not only radiative (e.g., fluorescence), but also mixed: radiative and non-radiative (or radiationless). During non-radiative relaxation, the excited states are converted to ground states at so-called conical intersection points on the potential energy hyper-surface. In the case of benzene monomers, these CI points have been carefully mapped and now it is known exactly under which energetic and geometrical conditions [13,14,15,16,17,18,20] the S${}_{1}$ excited state can non-radiatively flip to the S${}_{0}$ ground state. These transitions require the aromatic planar structure of the benzene molecule to be broken and a half-boat geometry to form [18]. The question arises as to how, when benzene molecules are organized into dimer configurations, at least one of the monomers can be distorted to form a CI geometry and thus relax radiationless. Accordingly, a dimer geometry in which one of the monomers showed the half-boat and the other one exhibited the aromatic planar monomer structure was assembled. This dimer geometry was then used as an initial geometry for the optimization of the CI points. The optimization was performed using the SF-TDDFT method, where the same XC functional and basis set (ωB97X-D3/ma-def2-TZVPP) was considered for the finding of the equilibrium geometry. During the geometry search, depending on which side of the half-boat geometry was oriented relative to the other planar monomer ring, two CI geometries were identified. For their molecular graphics see Figure 2a,b, while their geometrical parameters are given in the 5th and 6th double rows of Table 1. Analyzing the geometric parameters, it can be observed that both the half-boat and the aromatic ring structure are essentially preserved within the two dimer structures. What changes significantly is the relative position of the two monomers within the dimer compared to the excimer configuration. This means that the monomers move away from each other, due to the half-boat distortion. The shortest C···C${}^{\prime }$ bond distances are 3.76 for CI${}_{M}$(I) and 3.60 Å for CI${}_{M}$(II), respectively. The conformational energy also changes significantly compared to the R${}_{e}$(S${}_{1}$) excimer energy. It is 17.46 kcal/mol for CI${}_{M}$(I) and 17.60 kcal/mol for CI${}_{M}$(II). This happens, of course, not only because the half-boat monomer has a relatively high distortion energy, but also because the intermolecular interaction energy between the monomers is reduced. The calculated conformational energy difference between the equilibrium S${}_{1}$ and the half-boat CI geometries for the isolated monomer is 8.30 kcal/mol, calculated at the same level of theory as in the dimer case. Of course, it is almost impossible to estimate exactly how much of the conformational energy difference is due to intermolecular energy changes and how much is due to geometric deformations of the individual dimers, but by comparing the lengths of the C–C bonds for the different cases we can get a realistic view of which has the greatest influence on these changes. A comparison of bond lengths can be found in Appendix A in the Appendix A.

In the case of catechol, it has already been observed [34] that there are not only CI geometries where only one monomer is distorted, but also those where both monomers show some kind of change with respect to the equilibrium dimer geometry of the S${}_{1}$ excited state. A similar phenomenon was also observed experimentally in the case of much more complex molecular aggregates [48]. In the present case, two such dimer-type initial geometries have been assembled. Depending on whether the two monomers’ aromatic rings were facing each other or facing in opposite directions, the conformations were called *parallel* and *antiparallel*, respectively. The CI geometry search was performed considering the same SF-TDDFT method and applying the same ωB97X-D3/ma-def2-TZVPP computation scheme. The molecular graphics of the two dimer-type CI geometries, called CI${}_{D}$(I) and CI${}_{D}$(II), are shown in Figure 2c,d, while their geometrical parameters are given in the 7th and 8th double rows of Table 1. Analyzing the *parallel* CI geometry, it is important to note that the benzene rings originally facing each other are rotated during the CI optimization, and the dihedral angle C${}_{6}$–C${}_{3}$–C${}_{3}^{\prime }$–C${}_{6}^{\prime }$ becomes 31.6° (See Figure 2c). Furthermore, it can be observed that, in contrast to the CI${}_{M}$(I or II) monomer-type CI geometries, in this case both monomers are deformed with respect to the planar ring structure, as was observed for either the R${}_{e}$(S${}_{0}$) or R${}_{e}$(S${}_{1}$) equilibrium geometries. Although this deformation is not the same for the monomers, the changes in the length of the carbon–carbon bonds within the monomers do not differ greatly when the corresponding bonds are compared between the two monomers. The largest differences are seen between C${}_{2}$–C${}_{3}$ and C${}_{2}^{\prime }$–C${}_{3}^{\prime }$ as well as between C${}_{4}$–C${}_{5}$ and C${}_{4}^{\prime }$–C${}_{5}^{\prime }$ bonds, which are still only 0.018 Å. What is different from the monomer-type CI geometries is that a new bond is formed between C${}_{3}$ and C${}_{3}^{\prime }$ atoms with a bond length of 1.84 Å and a BOI of 0.32 and 0.23 when considering the ground-state and the S${}_{1}$ excited-state electron densities, respectively. If the energetic aspects are taken into account, it can be seen that the CI${}_{D}$(I) geometry gives only a 1.18 kcal/mol higher value than the R${}_{e}$(S${}_{1}$) equilibrium geometry. This conformational energy value is much lower than that obtained for CI${}_{M}$(I, or II) monomer-type CI geometries. As far as the *antiparallel* dimer-type CI geometry is concerned, a more and a less distorted monomer geometry is observed, also connected by a C–C${}^{\prime }$ bond. Here, one monomer keeps the planarity of the carbon atoms, only the aromatic structure changes, while the other monomer adopts a form close to the well-known half-boat configuration (See Figure 2d). However, it is not in the form of prefulvene as in the case of benzene monomer’s CI geometry, but more similar to cyclohexane’s boat form. The relative orientation that actually determines the antiparallel direction is the dihedral angle C${}_{6}$–C${}_{3}$–C${}_{3}^{\prime }$–C${}_{6}^{\prime }$, which in this case is 178°, whose value proves the antiparallel nature of the configuration. However, for CI${}_{D}$(II), the C${}_{3}$–C${}_{3}^{\prime }$ bond is much shorter than for CI${}_{D}$(I). Now this value is 1.59 Å and the BOIs are 0.75 and 0.72 depending on whether the ground-state or excited-state electron density is used. Based on the BOI values, these bonds are quite close to the classical C–C simple bonds. CI${}_{D}$(II), similar to CI${}_{D}$(I), also shows a small energy difference of 1.00 kcal/mol compared to the equilibrium geometry R${}_{e}$(S${}_{1}$).

To ensure the validity of the CI${}_{D}$(I) and CI${}_{D}$(II) geometries, a similar CI search was performed using a multiconfigurational CASSCF method. Accordingly, an active space of 11 molecular orbitals (six of them are occupied and five are unoccupied) and 12 electrons were set. For the selected molecular orbitals, see the MOs listed in the Appendix A at L2 for CI${}_{D}$(I) and at L3 for CI${}_{D}$(II). The CASSCF result fully confirms the validity of the dimer-type CI geometries obtained by the SF-TDDFT method. The molecular graphics of the two dimer-type CI geometries are shown in Figure 3a,b, while their geometrical parameters are given in the 12th and 13th double rows of Table 1. More significant differences between the CASSCF and SF-TDDFT results were found for the CI${}_{D}$(I) geometry, where CASSCF tends to form two C–C${}^{\prime }$ bonds between the two monomers, but with longer bond lengths (2.010 Å and 2.049 Å, respectively) and the monomers are less rotated with respect to each other (only 18.7° compared with 31.6° in the case of SF-TDDFT). This difference can also be explained by the fact that dispersion-type electron correlation effects are not taken into account in the CASSCF method. For the CI${}_{D}$(II) geometry, where there is no overlap between the two monomers and hence no significant dispersion effects, the agreement between CASSCF and SF-TDDFT results is much better. This is clearly demonstrated when comparing the length of the carbon–carbon bonds within the monomers, but it is particularly interesting that the length of the C–C${}^{\prime }$ vertical bond is also barely different between the two methods (1.546 Å vs. 1.585 Å).

It would also be worth checking whether it is possible to find other CI points in addition to the two conformations *parallel* and *antiparallel* CI${}_{D}$s. In order to avoid falling in one of the two CI points, a constrained geometry CI point search was performed, where the two monomer geometries were rotated by 60 and 100 degrees along the C–C${}^{\prime }$ bond and this value was fixed during the CI search. Hence, two additional CI geometries were identified whose geometric parameters are given in rows 9 and 10 of Table 1, called CI${}_{D}$(60°) and CI${}_{D}$(100°), respectively. However, these geometries are more difficult to access from an energetic point of view, as they have a higher value compared with the R${}_{e}$(S${}_{1}$) geometry than that for CI${}_{D}$(I) and CI${}_{D}$(II) points. Their conformational energy values are 14.32 kcal/mol and 6.98 kcal/mol, respectively, while their shortest C–C${}^{\prime }$ vertical bond distances are 1.753 Å and 1.930 Å.

### 2.3. Possible Relaxation Scenarios

For monomers, which essentially means the gas phase or the dilute solution in the liquid phase, the relaxation of excited states is achieved in a way already well known from the literature [13,14,15,16,17,18,49,50,51,52,53]. It raises the question of how these CI geometries can be achieved and what happens to the dimer system after it loses its excited state without radiation. What clearly emerges from the present analyses so far is that the dimer structure inhibits the formation of monomer-type CI geometries at the S${}_{1}$ excited-state level through the strong interaction between monomers more than would be the case for isolated monomers. In this sense, access to CI${}_{M}$(I, or II) geometries during the relaxation process is energetically less likely. Our analysis also shows that, for dimeric aggregation, dimer-type CI geometries can be achieved with a much smaller energy investment, which is not only due to the smaller change in intermolecular interaction energy, but should also be aided by the presence of a smaller deformation energy of the monomers within the dimer system. However, different preconditions must be met for the formation of CI${}_{D}$(I) and CI${}_{D}$(II) geometries. For the occurrence of CI${}_{D}$(I), the formation of the excimer geometry is likely to be important, as it is necessary for the monomers to have a face-to-face configuration, whereas for CI${}_{D}$(II) the existence of the excimer geometry is not necessary. At the same time, it has been shown by Subramaniam Iyer et al. [26] that, in the condensed phase, the inhibited movement of monomers, together with the redistribution of vibrational energy along multiple pathways, allows for a longer lifetime of the excimer (tens of ps), which in turn increases the likelihood of the formation of CI${}_{D}$(I) geometry.

The probability of the system reaching the CI${}_{D}$(I or II) points starting from the equilibrium geometry R${}_{e}$(S${}_{1}$) depends on the path of the dimer configuration along the potential energy surface. Accordingly, by applying the NEB method, the energetically optimal paths were calculated for both CI${}_{D}$(I) and CI${}_{D}$(II). It was found that for the point CI${}_{D}$(I) there exists a transition state (TS${}_{D}$(I)) with energy higher than the energy of the endpoints (R${}_{e}$(S${}_{1}$) and CI${}_{D}$(I)). Its molecular graph is shown in Figure 4, while its geometric parameters are given in row 11 of Table 1. Compared to the equilibrium geometry of R${}_{e}$(S${}_{1}$), TS${}_{D}$(I) has a 7.54 kcal/mol higher energy and the two monomers have fairly similar deformations, where the shortest C–C bond distance is 2.301 Å. This energy difference resulting from the TS${}_{D}$(I) geometry is not so large that the system would not be able to turn from the excimer state to the CI${}_{D}$(I) geometry, i.e., a relaxation path through the CI${}_{D}$(I) point cannot be excluded. As for reaching the CI${}_{D}$(II) point, the NEB method did not find any critical (local minimum or maximum) point, which means that from the R${}_{e}$(S${}_{1}$) equilibrium dimer geometry the dimer system is continuously rising in energy until it reaches the CI${}_{D}$(II) point. Summarizing our results thus far, from an energetic point of view, reaching the CI${}_{D}$(II) point and relaxing to the ground-state electronic level seems even more probable than in the previous CI${}_{D}$(I) case, since there is no need to pass through a higher potential barrier than that defined by the CI${}_{D}$(II) energy level itself.

As a next step, it is also important to investigate how the geometries returned in the ground state relax through CI points towards the final products. Since in the cases of the geometries CI${}_{D}$(I) and CI${}_{D}$(II) vertical C–C${}^{\prime }$ bonds are formed, it is not always the rule that they have to disappear in the ground state. Experimental study has shown [28] that covalent C–C bonds between the two monomers can be created and the cyclobutane benzene dimer molecule can be formed. Accordingly, geometry optimization for the cyclobutane benzene dimer with both the *syn*- and *anti*-addition forms was performed considering the same ωB97X-D3/ma-def2-TZVPP calculation setup. The molecular graphics of the two addition forms, called R${}_{e}^{CAo}$ and R${}_{e}^{CAa}$, are shown in Figure 5a,b, while their geometrical parameters are given in the 2nd and 3rd double rows of Table 1. As can be observed, the *anti*-addition is approximately 0.1 eV more stable energetically than the *syn*-addition conformation. Starting from the CI${}_{D}$(I) geometry, the relaxation channel can theoretically continue in two directions: (i) since in the ground state this dimer distance is smaller than the stacking distance in the equilibrium geometry, the two benzene dimers will push each other and thus the monomers will move away from each other; (ii) since the second C–C${}^{\prime }$ distance is only 2.104 Å (compared with the shortest 1.844 Å), it is also possible that the cyclo addition configuration will be formed leading to the *syn*-addition geometry (see Figure 5a) during relaxation. If and when this second case occurs, the geometry of R${}_{e}^{CAa}$ does not remain stable, easily decaying back to the monomeric form of the two benzenes, as was shown in Ref. [28]. In the case of the relaxation from CI${}_{D}$(II), we do not see any possibility for a second C–C${}^{\prime }$ bond to form the R${}_{e}^{CAo}$ (or *anti*-addition) geometry (see Figure 5b), only the separation of the two monomers and the reconstruction of their aromatic structures is possible. It can be concluded that, for both CI${}_{D}$(I) and CI${}_{D}$(II) geometries, relaxation will mainly lead back to the individual monomer forms, while the formation of other types of photoproducts is very unlikely.

## 3. Materials and Methods

The benzene dimer equilibrium configuration geometry at the ground-state electronic level was carried out in the framework of the density functional theory (DFT) considering the ωB97X exchange-correlation (XC) functional [54] combined with the D3-type empirical dispersion correction scheme [55,56] and applying the minimally augmented [57] ma-def2-TZVPP triple-ζ basis set of the Karlsruhe group [58], as implemented in the Orca program suite [59,60]. The equilibrium geometry of the benzene dimer computed at the first electronic excited-state level was obtained using the time-dependent DFT method including the Tamm–Dancoff approximation (TDA) [61]. The RIJCOSX approximation [62] developed to accelerate Hartree–Fock and hybrid DFT calculations was considered together with the Def2/J [63] auxiliary basis set for Coulomb fitting and the def2-TZVPP/C [64] auxiliary basis set for correlation fitting in the case of TDDFT calculations. The S${}_{0}$/S${}_{1}$ CI geometries were computed considering the spin-flip TDDFT (or SF-TDDFT) method [65,66,67,68] implemented in the same Orca package. The multiconfigurational CASSCF results were obtained using the Molpro program package [69,70,71], where instead of the ma-def2-TZVPP the 6–31G(d,p) basis set was considered. The nudged elastic band (NEB) method [72,73] was used to locate transition state (TS) geometries and to find minimum energy paths (MEPs) connecting different typical geometries, such as equilibrium and CI points on the potential energy surface. The bond order indexes were computed considering the Mayer’s indexing scheme [74,75]. The molecular geometries were built, analyzed and further manipulated using Gabedit [76] and Avogadro [77] programs, while the molecular graphics were created using the GaussView5.0.9 [78] software.

## 4. Conclusions

In this work, the equilibrium geometries of the ground and first electronic excited states as well as the radiationless deactivation channels of benzene in its monomer and dimer configuration were investigated using the standard linear-response and spin-flipped TDDFT methods as well as by CASSCF methods considering the minimally augmented def2-TZVPP and the 6–31G(d,p) basis sets. The equilibrium geometry of the first excited state of the dimer system, computed at the ωB97X-D3/ma-def2-TZVPP level of theory, exhibits a more compact stacking (so-called excimer) configuration, where the ground-state stacking distance decreases by approximately 0.5 Å and the dimer dissociation energy increases significantly (up to −14.00 kcal/mol). Furthermore, two families of conical intersection geometries, computed at the SF-ωB97X-D3/ma-def2-TZVPP level of theory, have been identified and named monomer- and dimer-type CI configurations, respectively. The monomer-type CI is characterized by the fact that only one monomer undergoes distortion, while the other monomer keeps the well-known ground-state aromatic ring structure, but the interaction between the two monomers is significantly decreased. However, in the case of dimer-type CI geometries, an additional C–C bond appears between one carbon atom of each of the two monomers, but the smaller geometric distortion of the monomers and the energy surplus created by the new bonding make these geometries more accessible than monomer-type CIs. The validity of the dimer-type CI geometries obtained by SF-TDDFT was also confirmed at the CASSCF(12,11)/6–31G(d,p) level of theory. Concerning the relaxation channels, it has been shown that the radiationless decay to the ground-state electron configuration of the benzene dimer through both CI${}_{D}$(I) and CI${}_{D}$(II) geometries is possible, even if the passing of an intermediate barrier is required for CI${}_{D}$(I). Overall, it can be said that, in the case of molecular aggregations, in addition to the well-known relaxations through CI points of the individual molecules, relaxations through intermolecular-type CIs can also help benzene to reach the ground electronic state radiationless.

## Figures and Tables

**Figure 1 ijms-24-02906-f001:**
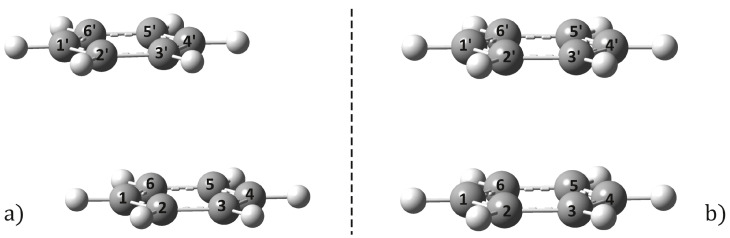
Equilibrium geometry structures of benzene dimer computed for ground (**a**) and first electronic excited (**b**) state considering the ωB97X-D3/ma-def2-TZVPP computation scheme at DFT and TDDFT levels of theory, respectively.

**Figure 2 ijms-24-02906-f002:**
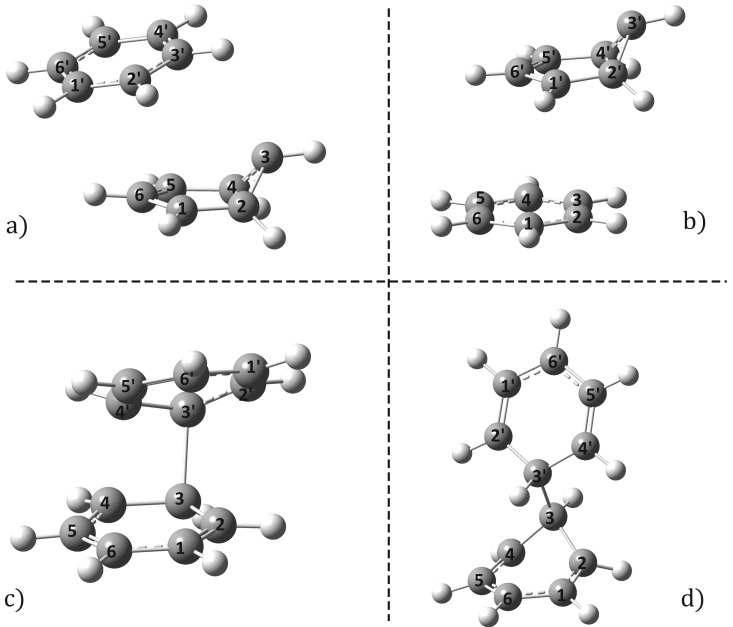
Conical intersection geometries of benzene dimer for monomer- ((**a**) CI${}_{M}$(I) and ((**b**) CI${}_{M}$(II)) as well as for dimer-type ((**c**) CI${}_{D}$(I) and ((**d**) CI${}_{D}$(II)) configurations computed at the spin-flipped TDDFT level of theory by considering the ωB97X-D3/ma-def2-TZVPP computation scheme.

**Figure 3 ijms-24-02906-f003:**
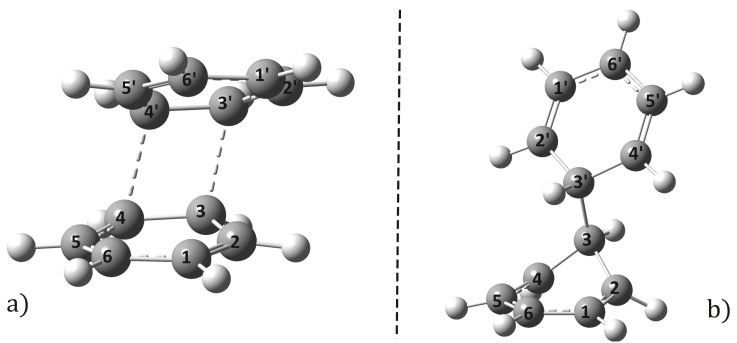
The two ((**a**) *parallel* and (**b**) *antiparallel*) dimer-type conical intersection geometries of benzene dimer computed at CASSCF(12,11)/6–31G(d,p) level of theory.

**Figure 4 ijms-24-02906-f004:**
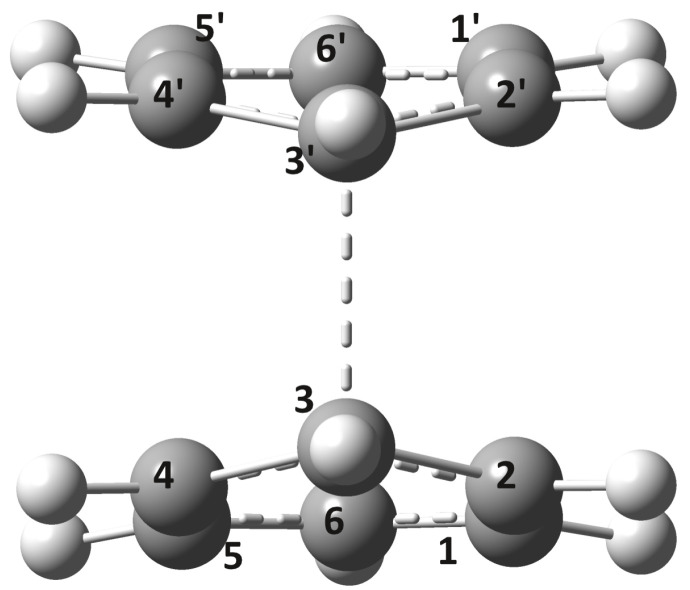
The transition state (TS${}_{D}$(I)) configuration between the R${}_{e}$(S${}_{1}$) and CI${}_{D}$(II) geometries computed at the NEB-TS-ωB97X-D3/ma-def2-TZVPP level of theory.

**Figure 5 ijms-24-02906-f005:**
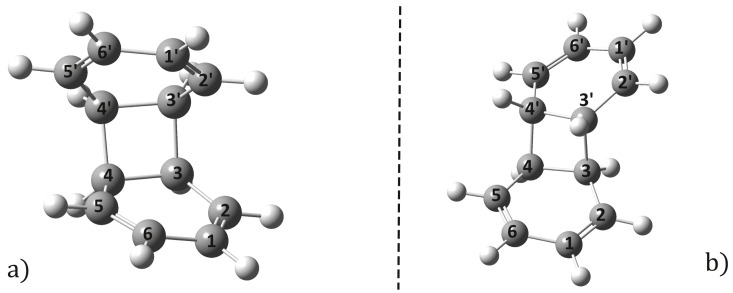
The cyclobutane benzene dimer geometry with *syn*- (**a**) and *anti*- (**b**) conformations computed at the DFT- ωB97X-D3/ma-def2-TZVPP level of theory.

**Table 1 ijms-24-02906-t001:** The bond lengths between the carbon atoms (in Å) and the conformational energy differences (last column) considering the first excited-state equilibrium structure as reference geometry (in eV and in kcal/mol in parenthesis) of benzene dimer computed for different equilibrium (R${}_{e}$) and S${}_{0}$/S${}_{1}$ conical intersection (CI) geometries at DFT, TDDFT, SF-TDDFT and CASSCF levels of theory. The XYZ coordinates of each geometry structure are listed (L1) in the Appendix A.

Geom.	Benzene Monomer’s C–C Bonds						Dimer	Energy
	C${}_{1}$–C${}_{2}$	C${}_{2}$–C${}_{3}$	C${}_{3}$–C${}_{4}$	C${}_{4}$–C${}_{5}$	C${}_{5}$–C${}_{6}$	C${}_{1}$–C${}_{6}$	C···C${}^{\prime }$	ref:
	C${}_{1}^{\prime }$–C${}_{2}^{\prime }$	C${}_{2}^{\prime }$–C${}_{3}^{\prime }$	C${}_{3}^{\prime }$–C${}_{4}^{\prime }$	C${}_{4}^{\prime }$–C${}_{5}^{\prime }$	C${}_{5}^{\prime }$–C${}_{6}^{\prime }$	C${}_{1}^{\prime }$–C${}_{6}^{\prime }$		R${}_{e}$(S${}_{1}$)
**DFT:** R${}_{e}$(S${}_{0}$), R${}_{e}^{CAo}$(S${}_{0}$), R${}_{e}^{CAa}$(S${}_{0}$), TS${}_{D}$(I); **TDDFT:** R${}_{e}$(S${}_{1}$); **SF-TDDFT:** CI${}_{M}$(I or II), CI${}_{D}$(I or II)								
(using the ωB97X-D3/ma-def2-TZVPP computation scheme)								
R${}_{e}$(S${}_{0}$)	1.387	1.387	1.387	1.387	1.387	1.387	3.579	−5.089
	1.387	1.387	1.387	1.387	1.387	1.387		
R${}_{e}^{CAo}$(S${}_{0}$)	1.330	1.497	1.553	1.495	1.468	1.387	1.563	−2.840
	1.330	1.495	1.554	1.498	1.330	1.468	1.564	
R${}_{e}^{CAa}$(S${}_{0}$)	1.331	1.493	1.554	1.497	1.331	1.468	1.562	−2.938
	1.331	1.493	1.554	1.496	1.331	1.468	1.554	
R${}_{e}$(S${}_{1}$)	1.401	1.401	1.401	1.401	1.401	1.401	3.073	0.000
	1.401	1.401	1.401	1.401	1.401	1.401		
CI${}_{M}$(I)	1.454	1.440	1.440	1.454	1.381	1.381	3.764	0.757
	1.388	1.387	1.387	1.386	1.388	1.387		(17.46)
CI${}_{M}$(II)	1.387	1.387	1.387	1.386	1.387	1.387	3.602	0.763
	1.458	1.437	1.440	1.451	1.384	1.378		(17.60)
CI${}_{D}$(I)	1.363	1.449	1.492	1.400	1.371	1.418	1.844	0.051
	1.359	1.431	1.486	1.418	1.369	1.427		(1.18)
CI${}_{D}$(II)	1.406	1.482	1.479	1.407	1.397	1.399	1.585	0.043
	1.359	1.482	1.482	1.359	1.408	1.408		(1.00)
CI${}_{D}$(60°)	1.366	1.464	1.541	1.407	1.374	1.408	1.753	0.621
	1.364	1.428	1.457	1.413	1.373	1.433		(14.32)
CI${}_{D}$(100°)	1.415	1.489	1.435	1.370	1.418	1.370	1.930	0.302
	1.369	1.420	1.481	1.429	1.364	1.422		(6.98)
TS${}_{D}$(I)	1.366	1.439	1.438	1.365	1.408	1.408	2.301	0.327
	1.365	1.438	1.439	1.365	1.408	1.408		(7.54)
CASSCF(12,11)/6–31G(d,p)								
CI${}_{D}$(I)	1.358	1.466	1.479	1.426	1.363	1.446	2.010	
	1.364	1.426	1.481	1.452	1.346	1.445	2.049	
CI${}_{D}$(II)	1.446	1.497	1.492	1.407	1.397	1.399	1.546	
	1.365	1.509	1.509	1.366	1.424	1.425		

M: *monomer*, D: *dimer*, C*_i_*–C*_j_* belongs to monomer A, C${}_{i}^{\prime }$–C${}_{j}^{\prime }$ belongs to monomer B.

## Data Availability

Data are available in a publicly accessible repository.

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
