# Peer review of "Intermolecular-Type Conical Intersections in Benzene Dimer"

_ijms, 2023, doi:10.3390/ijms24032906_

Round 1

Reviewer 1 Report

The article is a result of an interesting case study of intermolecular-type conical intersections in benzene dimer. The authors give a detailed description of the possible photo-induced radiationless decay channels in benzene dimer over the newly proposed dimer-type conical intersection points. The calculations were carried out using the ORCA and MOLPRO program packages and the well-known softwares for molecular graphics. The article is well organized, the graphical presentation is clear, and the results seem compelling. Therefore, I recommend publishing in IJMS in its current form with minor corrections:

1.     Please, clarify the terms “dispersion correction” (line 150) and “dispersion electron correlation effect” (line 153). Does it mean that you discriminate the origin of dispersion forces and the effects of electron correlation?

2.     Line 136: should be “remain equivalent”?

3.     Line 163: should be “exactly known”?

Author Response

- The authors thank the Referee for pointing out the paper weaknesses and for his (her) helpful comments on the manuscript. All comments are justified, thus we introduced additional information in our manuscript accordingly to Referee’s suggestion.

- Please, clarify the terms “dispersion correction” (line 150) and “dispersion electron correlation effect” (line 153). Does it mean that you discriminate the origin of dispersion forces and the effects of electron correlation?

- Answer: No, definitely not. But while the dispersion energy contribution is a pure electron correlation effect, in practice it has been estimated by an empirical correction (the Grimme D3 method). The possible misunderstanding may have arisen from this.

- Line 136: should be “remain equivalent”?

- Line 163: should be “exactly known”?

 - Answer: The grammar errors were corrected. See lines 144 and 174 in the revised manuscript.

Reviewer 2 Report

Summary

The authors of the paper entitled Intermolecular-type conical intersections in benzene dimer address the efficient radiationless relaxation from the first singlet excited state (S1) to the singlet ground state (S0) through conical intersections (CIs) for the benzene dimer. The authors use spin-flip time-dependent density functional theory (SF-TDDFT) and multiconfigurational method, namely CASSCF, to identify novel CI geometries that may lead to a more efficient internal conversion from S1 to S0.

General concept comments

The authors cleverly identify a clear gap in the knowledge of the relaxation of the benzene dimer from S1 to S0. As the authors pinpoint there are several studies of the excited dynamics of the benzene molecule, but little is known about the radiationless relaxation of the benzene dimer. The manuscript is concise and well-organised; the computational methods selected by the authors fulfil the requirements to address complex situations such as the optimisation of molecular structures at CIs. However, the manuscript contains some flaws that require correction.

The authors do not provide the geometries of the structures presented in the manuscript. Along the same line, they do not show the orbitals included in the active space for the CASSCF calculations. I do encourage the authors to include this information either in the main text or in the Supplementary Information section to ensure that the results are reproducible.

In Section 3.2 the authors carry out CASSCF calculations to validate the results obtained within SF-TDDFT using the exchange-correlation functional wB9X-D3, which partially accounts for dynamic correlation effects. The wavefunction obtained with CASSCF does not include dynamic correlation effects, so thus the results obtained with SF-TDDFT and CASSCF may agree for the wrong reasons. Did the authors carry out NEVPT2 or CASPT2 calculations to include dynamic correlation effects in the CASSCF wavefunction?

In Table 1 and in the caption of the same table, it is not clear which results have been obtained using DFT, TDDFT and SF-TDDFT. Could the authors specify which numbers have been obtained with DFT, which with TDDFT and which with SF-TDDFT?

Figures 1, 2, 3 and 4. Could the authors indicate on the top of each atom the corresponding atomic label: C1, C2, C'1, etc? Otherwise, it is rather difficult to follow the discussion of the results.

Specific comments

The authors refer to the CASSCF method as multireference. This is wrong. The method CASSCF is multiconfigurational, not multireference. 

Line 32: "...S1/S0 radiationless relaxation ..." I think that the authors mean radiative relaxation.

Line 32: "... theoretical and experimental methods." Reference missing after "methods."

Line 80: "... larger molecules require a less computationally intensive approach where the advantages provided by the DFT method can be exploited." I would ask the authors to rephrase this sentence. Large molecules do require demanding computational approaches, but state-of-art quantum chemistry methods cannot deal with large molecules including hundreds or thousands of atoms.

Line 148. Could the authors develop the concept of electron-hole interaction?

Line 178: "..., it can be observed that both the half-boat and the aromatic ring structure are essentially preserved within the two dimer structures." This is quite hard to see from Table 1. I do encourage the authors to include the bond distances in Figure 2.

Line 185: Are the monomer distortion and the intermolecular distance between monomers in CIM(I) and CIM(II) related?

Line 216: "This value is much lower than ...". What value do the authors refer to?

Line 226: "0.75 and 0.72 depending on whether the ground state or excited state electron density is used." Why do the authors use two different densities?

Line 309: The authors claim that the anti-addition and the syn-addition conformations are almost identical because they differ in 0.1 eV. I disagree with the authors. 0.1 eV is not a small energy difference, especially considering internal conversion processes.

Line 302: Why do the authors refer to the cyclobutane benzene dimer? Is it some special nomenclature?

Line 392: The title of reference 15  is missing.

To finish, the article is full of spelling and grammar mistakes. I gently ask the authors to put more attention to the writing.

Author Response

- The authors thank the Referee for pointing out the paper weaknesses and for his (her) helpful comments on the manuscript. All comments are justified, thus we introduced additional information in our manuscript accordingly to Referee’s suggestion.

- The authors do not provide the geometries of the structures presented in the manuscript. Along the same line, they do not show the orbitals included in the active space for the CASSCF calculations. I do encourage the authors to include this information either in the main text or in the Supplementary Information section to ensure that the results are reproducible.

- Answer: The XYZ coordinates for each geometry included in Table 1 were included in the Supplementary Material file.

- In Section 3.2 the authors carry out CASSCF calculations to validate the results obtained within SF-TDDFT using the exchange-correlation functional wB9X-D3, which partially accounts for dynamic correlation effects. The wavefunction obtained with CASSCF does not include dynamic correlation effects, so thus the results obtained with SF-TDDFT and CASSCF may agree for the wrong reasons. Did the authors carry out NEVPT2 or CASPT2 calculations to include dynamic correlation effects in the CASSCF wavefunction?

 - Answer: The Reviewer is absolutely right. In our case, the CASSCF is indeed not comparable with the SF-TDDFT results. We only used the CASSCF calculations to confirm the possible existence of CI geometries obtained by the SF-TDDFT method, which is still not widely accepted, and therefore we wanted to be able to confirm the realistic existence of these CI geometries. However, as already mentioned in the manuscript, for the anti-parallel geometry the dynamic correlation effects do not play such a significant role anymore (no stacking configuration between the rings) and it is observed that the CASSCF and SF-TDDFT results are in much better agreement. As far as the NEVPT2 or CASPT2 method is concerned, it could indeed be a solution that would better characterize these conical intersection geometries. In this case, software development would be needed that includes gradients and CI search modules at the level of these theories. Such a solution is perhaps only offered by the BAGEL software developed by the Shiozaki group, but it is certainly a serious challenge both scientifically and in terms of computational resources, which would be the subject of a separate article.

- In Table 1 and in the caption of the same table, it is not clear which results have been obtained using DFT, TDDFT and SF-TDDFT. Could the authors specify which numbers have been obtained with DFT, which with TDDFT and which with SF-TDDFT?:

- Answer: In Table 1, the names of the geometries obtained by the respective methods are marked separately. See the third row of the table.

- Figures 1, 2, 3 and 4. Could the authors indicate on the top of each atom the corresponding atomic label: C1, C2, C'1, etc? Otherwise, it is rather difficult to follow the discussion of the results.

- Answer: Carbon atoms were labeled with the corresponding atomic label in each figure.

- The authors refer to the CASSCF method as multireference. This is wrong. The method CASSCF is multiconfigurational, not multireference.

- Answer: Thank you for pointing out this confusion. The “multireference” notation was changed to “multiconfigurational” expression.

- Line 32: "...S1/S0 radiationless relaxation ..." I think that the authors mean radiative relaxation. See line 35 in the revised manuscript.

- Answer: We apologize for this incorrect wording. The sentence was corrected;

- Line 32: "... theoretical and experimental methods." Reference missing after "methods."

- Answer: The corresponding bibliographic references were added. See line 34 in the revised manuscript.

- Line 80: "... larger molecules require a less computationally intensive approach where the advantages provided by the DFT method can be exploited." I would ask the authors to rephrase this sentence. Large molecules do require demanding computational approaches, but state-of-art quantum chemistry methods cannot deal with large molecules including hundreds or thousands of atoms.

- Answer: The corresponding text was rephrased. See lines 82-84 in the revised manuscript.

- Line 148. Could the authors develop the concept of electron-hole interaction?

- Answer: The concept of electron-hole interaction as a possible explanation for the increase in intermolecular interaction energy in the excited state is taken from Ref. 24, where the forces involved in the formation of the benzene state and the formation of the excimer and its electron structure were analyzed in detail. A short explanation text was also included in the manuscript, but in Ref. 24 these effects are described in more detail. See lines 157-159 in the revised manuscript.

- Line 178: "..., it can be observed that both the half-boat and the aromatic ring structure are essentially preserved within the two dimer structures." This is quite hard to see from Table 1. I do encourage the authors to include the bond distances in Figure 2.

- Answer: Given that carbon atoms have been indexed for each figure in the revised manuscript, we believe that it is now much easier to identify the bond lengths of the structures shown in the figures.

- Line 185: Are the monomer distortion and the intermolecular distance between monomers in CIM(I) and CIM(II) related?

- Answer: Is hard to demonstrate this. However, it is known from the theory of intermolecular interactions that within a dimer configuration, the electron structure of one monomer affects the electron structure of the other monomer and vice versa. From this point of view, the CI geometry of isolated benzene molecules can vary (to a very small extent) with respect to the monomer-type CI geometry within the dimer. Even if it is energetically very difficult to estimate this difference, a comparison can be made in terms of C-C bonds. A short text regarding this observation was added in the manuscript and a further table with the C-C bonds in the half-boat configuration for the isolated monomer as well as the CI_M(I) and CI_M(II) geometries was included in the Supplementary material file. See lines 199-204 in the revised manuscript.

- Line 216: "This value is much lower than ...". What value do the authors refer to?

- Answer: Sorry for not being more precise. It is about the energy difference (or conformational energy) between values obtained for the given CI geometries (CI_D(I) or CI_D(II)) and for the equilibrium S1 geometry. See line 230 in the revised manuscript.

- Line 226: "0.75 and 0.72 depending on whether the ground state or excited state electron density is used." Why do the authors use two different densities?

- Answer: Although for CI geometries, the energies of the two electronic states are almost equal, but this is obtained with two different densities. As a consequence, the Rho(S0) and Rho(S1) densities give different charge distributions and different bond orders. This is why we compute separately the bond orders for the two electronic states.

- Line 309: The authors claim that the anti-addition and the syn-addition conformations are almost identical because they differ in 0.1 eV. I disagree with the authors. 0.1 eV is not a small energy difference, especially considering internal conversion processes.

- Answer: Thank you for pointing out this misinterpretation. The corresponding text was changed.

- Line 302: Why do the authors refer to the cyclobutane benzene dimer? Is it some special nomenclature?

- Answer: Not, it isn’t. This description was taken from reference 28.

- Line 392: The title of reference 15 is missing.

- Answer: The title of Ref. 15 was corrected.

- To finish, the article is full of spelling and grammar mistakes. I gently ask the authors to put more attention to the writing.

- Answer: - The entire text has been checked for spelling and grammatical errors. If the Reviewer still considers that the revised manuscript needs major corrections, we will seek the help of a professional proofreader.

Reviewer 3 Report

In this manuscript, the authors analyzed conical intersection geometries in benzene dimers using SF-TDDFT and CASSCF methods. The authors need to address some concerns before it is considered for publication.

1. Please label the carbon atoms in the corresponding figures when the bond lengths and angles are mentioned.

2. The authors mentioned 6-31G** basis set in the abstract and conclusions, but it is not listed in the materials and methods.

3. What effects will the substitutional groups of the benzene dimers have on the conical intersection geometries?

4. It would be better to provide the combined files of optimized benzene dimer geometry as the supplementary material.

Author Response

- The authors thank the Referee for pointing out the paper weaknesses and for his (her) helpful comments on the manuscript. All comments are justified, thus we introduced additional information in our manuscript accordingly to Referee’s suggestion.

- Please label the carbon atoms in the corresponding figures when the bond lengths and angles are mentioned.

- Answer: Thank you for pointing out this shortcoming. For each figure, the carbon atoms have been indexed.

- The authors mentioned 6-31G** basis set in the abstract and conclusions, but it is not listed in the materials and methods.

 - Answer: Thank you for pointing out, again, this shortcoming. Details on the exact use of 6-31G** are given in the Materials and Methods section. See lines 111-112 in the revised manuscript.

- What effects will the substitutional groups of the benzene dimers have on the conical intersection geometries?

- Answer: Thank you very much, it is a very good question. Unfortunately, at this stage, we cannot give a precise answer. What we have observed in our studies, made both for benzene and catechol, is that the shape of the CI geometry depends on the nature of the side fragment and its position on the ring. We have included a remark on this in the manuscript for the sake of interest, but it is clear that further calculations are needed.

- It would be better to provide the combined files of optimized benzene dimer geometry as the supplementary material.

- Answer: The XYZ coordinates for each geometry included in Table 1 were included in the Supplementary Material file.

Round 2

Reviewer 2 Report

The authors have improved the quality of the article significantly. The article is easier to follow and they have clarified all my doubts. Before publication, I would like to suggest some minor amendments.

Specific comments

Line 40: "... this relaxation pathway is might given ..." --> "... this relaxation pathway might be given".

Line 75: "Furthermore, similar, intermolecular ..." --> "Furthermore, similar intermolecular-type CI ..."

Line 104: "The equilibrium geometry of benzene dimer computed at first electronic excited state level ...." --> "The geometry of the first electronic excited state of the benzene dimer was obtained ..."

Table 1: In the caption, the authors write TSD(II), but in the 11th row it says TSD(II).  According to the discussion in Section 3.3, there is no TSD(II) that links CIM(II) to CID(II). In the Supporting Information, the authors also refer to TSD(II) on page S6.

Line 131. A closing parenthesis is missing after Table 1.

Line 137: I suggest replacing the term "molecular graphics" used throughout the manuscript with "molecular structure".

Line 138: "... optimization at the S1 excited-state level" --> "... optimization of the S1 excited state".

Line 163: "What's more, ..." --> "What is more, ..."

Lines 199-204: The authors say that is not possible to estimate the contribution of the changes in the intermolecular energy and of the deformation of the monomer to the interaction energies of CIM(I) and CIM(II). I fully disagree with this statement. The authors could compute the deformation energy for each separate monomer to transform from the geometry they have in Re(S1) to the geometry they have in CIM(I) and CIM(II). Then, the authors could calculate the interaction energy of the deformed monomers. The former calculation would provide an estimate of the contribution due to the deformation of the monomers; the latter the contribution due to the intermolecular interaction.

Line 209: "aggregate" ---> "aggregates"

Line 246: "a multi-reference CASSCF method" --> "a multiconfigurational CASSCF method"

Line 259: The authors say: "For the CID(II) geometry, where there is no overlap between the two monomers and hence no significant dispersion effects, ...". I am not sure about this statement. A covalent bond binds the two monomers in CID(II), and thus the electron densities of the two monomers must overlap. In Figure L3 of the Supporting Information, the authors show that the HOMO (orbital 42) and the LUMO (orbital 43) are delocalized between the two monomers for CID(II). The agreement between the molecular structures obtained with SF-TDDFT using the functional wB97X-D3 that accounts for dispersion interactions, and the structures obtained using CASSCF shows that the active space selected for the CASSCF is sufficiently large to partially account for dispersion interactions.

Line 327: "...equilibrium geometry..." ---> It is not clear whether the authors mean the ground state equilibrium geometry of the CID(I) geometry.

Line 341: "radiation-less" --> radiationless

Author Response

- The authors thank the Referee for pointing out the paper weaknesses and for his (her) helpful comments on the manuscript. All comments are justified, thus we introduced additional information in our manuscript accordingly to Referee’s suggestion.

- Line 40: "... this relaxation pathway is might given ..." --> "... this relaxation pathway might be given".

- Answer: The corresponding sentence was corrected.

- Line 75: "Furthermore, similar, intermolecular ..." --> "Furthermore, similar intermolecular-type CI ..."

- Answer: The corresponding sentence was corrected.

- Line 104: "The equilibrium geometry of benzene dimer computed at first electronic excited state level ...." --> "The geometry of the first electronic excited state of the benzene dimer was obtained ..."

- Answer: The corresponding sentence was corrected.

- Table 1: In the caption, the authors write TSD(II), but in the 11th row it says TSD(II). According to the discussion in Section 3.3, there is no TSD(II) that links CIM(II) to CID(II). In the Supporting Information, the authors also refer to TSD(II) on page S6.

- Answer: The corresponding errors were corrected both in the manuscript and in the supplementary file.

- Line 131. A closing parenthesis is missing after Table 1.

- Answer: The corresponding sentence was corrected.

- Line 137: I suggest replacing the term "molecular graphics" used throughout the manuscript with "molecular structure".

- Answer: We apologize for this incorrect wording. The sentence was corrected;

- Line 138: "... optimization at the S1 excited-state level" --> "... optimization of the S1 excited state".

- Answer: The corresponding sentence was corrected.

- Line 163: "What's more, ..." --> "What is more, ..."

- Answer: The corresponding sentence was corrected.

- Lines 199-204: The authors say that is not possible to estimate the contribution of the changes in the intermolecular energy and of the deformation of the monomer to the interaction energies of CIM(I) and CIM(II). I fully disagree with this statement. The authors could compute the deformation energy for each separate monomer to transform from the geometry they have in Re(S1) to the geometry they have in CIM(I) and CIM(II). Then, the authors could calculate the interaction energy of the deformed monomers. The former calculation would provide an estimate of the contribution due to the deformation of the monomers; the latter the contribution due to the intermolecular interaction.

- Answer: Thank you very much for offering this solution. We will take into account in our further works.

- Line 209: "aggregate" ---> "aggregates"

- Answer: The corresponding sentence was corrected.

- Line 246: "a multi-reference CASSCF method" --> "a multiconfigurational CASSCF method".

- Answer: The corresponding sentence was corrected.

- Line 259: The authors say: "For the CID(II) geometry, where there is no overlap between the two monomers and hence no significant dispersion effects, ...". I am not sure about this statement. A covalent bond binds the two monomers in CID(II), and thus the electron densities of the two monomers must overlap. In Figure L3 of the Supporting Information, the authors show that the HOMO (orbital 42) and the LUMO (orbital 43) are delocalized between the two monomers for CID(II). The agreement between the molecular structures obtained with SF-TDDFT using the functional wB97X-D3 that accounts for dispersion interactions, and the structures obtained using CASSCF shows that the active space selected for the CASSCF is sufficiently large to partially account for dispersion interactions.

- Answer: It is hard to say that differences between the CASSCF and SF-TDDFT results seen in the case of CI_D(I) comes from the dispersion-type correlation effects or from other correlation effects which are included in the wB97X functional. It is important to note, that the empirically corrected dispersion does not cover the whole dispersion contribution, is just a correction of it fitted empirically. Referee is right that there is a substantial difference between the results obtained by the CASSCF and the wB97X XC functional, but as far as the empirical correction is concerned it depends on the sixth power of the distance between atom-pairs and is much smaller for CI_D(I), since the distances between atom-pairs are also larger.

- Line 327: "...equilibrium geometry..." ---> It is not clear whether the authors mean the ground state equilibrium geometry of the CID(I) geometry.

- Answer: The corresponding sentence was corrected.

- Line 341: "radiation-less" --> radiationless

- Answer: The corresponding sentence was corrected.
